# Appropriate Needle Length Determined by Ultrasonic Echography for Intramuscular Injection in Japanese Elderly over 50 Years

**DOI:** 10.3390/healthcare10050800

**Published:** 2022-04-25

**Authors:** Tetsuo Nakayama, Hisakuni Sekino, Hirokazu Aihara, Minoru Kino

**Affiliations:** 1Laboratory of Viral Infection, Ömura Satoshi Memorial Institute, Kitasato University, Minato-ku, Shirokane 5-9-1, Tokyo 108-8641, Japan; 2Sekino Hospital, Toshima-ku, Ikebukuro 3-28-3, Tokyo 171-0014, Japan; sekinoh@sekino-hospital.com; 3Shiki-Kashiwamachi Clinic, Kashiwa-cho 1-6-74, Shiki 353-0007, Japan; h-aihara@kawatsuru-g.jp; 4Department of Pediatrics, Osaka Asahi Children’s Hospital, Asahi-ku, Shinmori 4-13-17, Osaka 535-0022, Japan; kino@nakano-kodomo.or.jp

**Keywords:** intramuscular injection, muscle fascia, needle length, Japanese elderly

## Abstract

Adjuvanted vaccines are administered through intramuscular injection. To perform appropriate injection using an appropriate needle in different age groups or different daily living activities, we investigated the depth from the skin surface to muscle fascia and bone in the deltoid muscle area in 156 elderly aged ≥ 50 years by ultrasonic echography. Subjects consisted of 50 healthy elderly aged 50–64 years, 50 subjects aged 65–74 years, and 56 subjects aged ≥ 75 years (20 outpatients, 18 who needed nursing care, and 18 bedridden in a nursing home). The mean depth ± 1.0 SD from the skin surface to muscle fascia was 7.52 ± 2.13 mm for subjects aged ≥ 75 years, being shorter than 9.16 ± 3.02 mm in those aged 50–64years (*p* < 0.01). The depth from the skin surface to bone was 22.54 ± 3.85 mm for subjects aged ≥ 75 years and 25.41 ± 4.24 mm for those aged 65–74 years, significantly shorter than those aged 50–64 years (*p* < 0.01), depending on the reduced muscle volume. The subcutaneous volume length was greater in females (8.29 ± 2.63 mm) than in males (5.62 ± 2.80 mm) aged 50–64 years (*p* < 0.01). A similar result was obtained in those aged 65–74 years, but there was no difference in the muscle volume length. Our study found that a five-eighths of an inch (16 mm) needle was an appropriate length for average-sized elderly aged ≥ 50 years, but it should be longer for those with large body sizes.

## 1. Introduction

The administration of adjuvanted vaccines to children is recommended through intramuscular injection worldwide, but they are administered through subcutaneous injection in Japan [1,2]. From the 1960s to the 1970s, it was noticed as medical malpractice that the administration of antibiotics together with antipyretics in the thigh lesion led the muscle contraction in young infants [3]. The reason for the contraction was the extremely high osmotic pressure and pH, different from the physiological condition [3,4]. Although there was no case of muscle contracture after the administration of vaccines in children, vaccines have been administered subcutaneously ever since intramuscular injection was prohibited in 1976. We reported that there was no evidence of muscle necrosis or fibrosis after the administration of adjuvanted and nonadjuvanted vaccines in mice. Aluminum-adjuvanted vaccines induced inflammatory nodules with the infiltration of inflammatory cells or macrophages at the injection sites [5]. Inflammatory nodules spread into muscle bundle spaces without the degeneration or atrophic changes in muscle tissues. Infiltrating cells were characterized initially as neutrophils and later as macrophages [5].

A newly introduced human papillomavirus vaccine (HPV) has been recommended to be administered intramuscularly since its introduction to Japanese female middle school students [6]. Several adverse reactions were reported after the introduction [7]. Some localized pains may be due to the unfamiliar procedure of intramuscular administration. Especially, recombinant purified zoster vaccine (Shingrix^®^, GlaxoSmithKline, Belgium) was licensed in 2019, demonstrating more than 95% clinical efficacy but high incidences of local pain, redness, and swelling [8]. In December 2019, COVID-19 occurred in Wuhan, China, and caused a pandemic. New-concept vaccines based on mRNA were developed, and immunization started in December 2020 in the U.S. [9]. High clinical efficacy was reported in the Phase III clinical trial, but there was a high incidence of local pain with systemic adverse events, such as fatigue, headache, and febrile illness. These new vaccines are introduced through intramuscular injection. Recombinant purified zoster vaccine is recommended for subjects aged ≥ 50 years, and COVID-19 mRNA vaccines initially for medical staff members and then for the elderly. General physicians in Japan are not accustomed to intramuscular immunization procedures. A proper injection procedure is required for immunization.

The appropriate needle length was determined as 25 mm for both males and females with body mass index (BMI) < 35, and 32 mm for those with BMI > 35 in Australia, using ultrasonic echography [10]. In addition, a 25 mm needle is recommended by the Centers for Disease Control and Prevention (CDC) as a standard procedure [11]. Several studies reported a length of 16 or 25 mm for large subjects in the U.S., Russia, and other counties [12]. Reactogenicity after immunization would be influenced by the immunization procedure with appropriate needle size and length. Subcutaneous and muscular volumes would differ depending on the racial differences and body size [13]. There is no evidence of an appropriate needle length for short-stature Japanese elderly aged ≥ 50 years. The objective of this study was to determine an appropriate needle length to achieve proper intramuscular injection to Japanese elderly with different nursing care needs, especially those aged ≥ 75 years, using ultrasonic echography.

## 2. Subjects and Methods

### 2.1. Subjects

The sample size was established by referring to previous reports [10,11] and our report on children [14]. The study participants were randomly enrolled in the outpatient clinics, and bedridden subjects ≥ 75 years accommodated in the nursing home. A total of 160 subjects were enrolled after explaining the study design and obtaining written informed consent. To investigate the appropriate needle length in different age groups or different daily living activities, they were divided into three groups: 50 subjects aged 50–64 years, 50 subjects aged 65–74 years, and 60 subjects aged ≥ 75 years. Males and females were equally recruited. Sixty subjects aged ≥ 75 years were classified into three subgroups depending on daily activities: 20 subjects in the outpatient clinic, 20 subjects who needed nursing care, and 20 subjects who were bedridden. Exclusion criteria were subjects with anomalies in upper limbs, peripheral nervous disorders, muscular illness of upper limbs, acute systematic illnesses, and serious chronic ailments. Two cases were excluded in each group of those aged ≥ 75 years with nursing care and those bedridden because of blurred echography, and so a total of 156 subjects were included (Table 1).

This sample size represents the average physique of the elderly and appropriate sampling. Subjects aged ≥ 75 years were divided into three subgroups: outpatients, required nursing care, and bedridden in a nursing home, with 20 subjects in each. In these groups, there was a small variation in the physique.

### 2.2. Study Design

Before the entry into the study, height and body weight were measured, and body mass index (BMI) was calculated. The circumference in the center of the deltoid muscle area was measured. The length of subcutaneous tissues was measured at the center of the deltoid muscle of the nondominant arm using ultrasonic echography. The study protocol was approved by the ethics committee of Kitasato Institute Hospital (approved No. 20038), adopting the Ethical Guideline for Medical and Health Research involving human subjects and the Declaration of Helsinki.

### 2.3. Ultrasonic Echography

Ultrasonic echography was performed in the center of the deltoid muscle using Aplio 400/500 with the linear probe PLT-704SBT (Toshiba Medical Systems, Tokyo, Japan) or Prosound SSD-α10 with the linear probe UST-5411 (Hitachi Aloka-Medical, Tokyo, Japan). The thickness of the epidermis and distance from the skin surface to muscle fascia and bone were measured based on the different intensities of echograms [4]. The probe was held at 90° to the skin surface in a relaxed position. Ultrasonic images were examined by a medical echogram specialist (co-author MK), and unclear blurred images were excluded from the analysis, as was done in the study to determine an appropriate needle length for young infants [14]. Typical echography is shown in Figure 1.

### 2.4. Statistical Analyses

Statistical analyses were performed using the SAS system and BellCurve (Social Survey Research Information Co., Ltd., Tokyo, Japan) and significance were determined by Welch’s *t*-test.

## 3. Results

### 3.1. Body Weight, Height, BMI, and Circumference of Upper Arm in Different Age Groups

Body sizes were measured at the enrollment in the study, and the results are shown in Table 2. Height was 164.34 ± 8.86 cm (mean ± 1.0 SD) in subjects aged 50–64 years, higher than that in those aged 65–74 years (*p* < 0.01) and is higher (*p* < 0.05) compared with that in those aged ≥ 75 years. The body weight was 73.32 *±* 20.44 kg (mean ± 1.0 SD) in those aged 50–64 years, being lighter in older generations in comparison with each group (*p* < 0.01). BMI was 26.96 ± 6.40 (mean ± 1.0 SD), and the circumference of the upper arm around the center of the deltoid was 32.09 ± 5.59 cm in those aged 50–64 years, being larger compared with those aged 65–74, with significant differences (*p* < 0.05). They were smaller or shorter in those aged ≥ 75 years (*p* < 0.01). In bedridden subjects in a nursing home aged ≥ 75 years, data on body weight, BMI, and circumference of the upper arm were the lowest (*p* < 0.01).

### 3.2. The Depth from Skin Surface to Muscle Fascia and Bone in Different Age Groups

Ultrasonic echogram was taken from all participants, and typical ultrasonic echograms are shown for different age groups: 72-year-old male, 75-year-old female outpatient, and 85-year-old female bedridden in a nursing home in Figure 1. The thickness of the epidermis, subcutaneous tissue, and muscle volume were measured at different ultrasonic intensities.

Depth from the skin surface to muscle fascia and bone in different age groups is shown in Table 3. The epidermis is thin at 1.93 ± 0.54 mm (mean ± 1.0 SD, *p* < 0.01) in those aged ≥ 75 years compared with those aged 50–64 years. Subcutaneous volume length was 5.58 ± 2.03 mm (mean ± 1.0 SD, *p* < 0.05) in those aged ≥ 75 years, being shorter than that in those aged 50–64 years, and the muscle volume length was 15.02 ± 3.15 mm (mean ± 1.0 SD, *p* < 0.01) in those aged ≥ 75 years and 16.95 ± 3.19 mm (mean ± 1.0 SD, *p* < 0.01) in those 65–74 years, being significantly shorter than 19.44 ± 4.44 mm in those aged 50–64 years.

The mean depth ± 1.0 SD from the skin surface to muscle fascia was 7.52 ± 2.13 mm for subjects aged ≥ 75 years and shorter than 9.16 ± 3.02 mm in those aged 50–64 years (*p* < 0.01). For those aged ≥ 75 years, it was 7.51 ± 1.72 mm for outpatient subjects, 7.53 ± 2.56 mm for those who needed nursing care, and 6.44 ± 2.05 mm for those bedridden in a nursing home. The depth from the skin surface to bone was 22.54 ± 3.85 mm for subjects aged ≥ 75 years, and 25.41 ± 4.24 mm for those aged 65–74 years, and significantly shorter than those aged 50–64 years (*p* < 0.01). It was significantly shorter at 19.47 ± 5.55 mm for those who were bedridden in a nursing home than 22.82 ± 4.06 mm for outpatients (*p* < 0.05).

### 3.3. The Depth from Skin Surface to Muscle Fascia and Bone in Different Genders

Gender differences in subcutaneous mass are shown in Table 4. The subcutaneous volume length was larger in females (8.29 ± 2.63 mm) than in males (5.62 ± 2.80 mm) aged 50–64 years (*p* < 0.01). Equivalent results were obtained in those aged 65–74 years; there was no difference in muscle volume length. Depth from the skin surface to muscle fascia was longer in females than in males in both, those aged 50–64 years and 65–74 years (*p* < 0.01).

Irrespective of age, body structure, and gender, a 16 mm needle reached the muscular mass, and a 25 mm needle reached the bone in some slim elderly subjects.

### 3.4. Correlation between Subcutaneous Mass and Body Structure (BMI and Circumference)

Correlations between the depth to the muscle surface or the bone and BMI or circumference of the deltoid muscle area were investigated, and the results are shown in Figure 2. Depth from the skin surface to muscle fascia was not correlated with BMI or circumference, but that to bone was correlated with the circumference around the deltoid area.

## 4. Discussion

Most vaccines are administered intramuscularly, and the immunogenicity and safety of vaccines are influenced by several factors: vaccine materials, components of adjuvant, and immunization procedures [15,16]. Vaccines are efficiently captured by antigen-presenting cells at the site of injection and are drained to the regional lymph nodes. Vaccine antigen transportation induces crucial priming of immune-competent cells [17]. Most vaccines are given to young infants and children intramuscularly because of their simplicity, and there is no significant difference in immunogenicity. It was reported that, depending on several kinds of vaccines given intramuscularly and subcutaneously, fewer episodes of severe local reactions were experienced in infants administered vaccines using a longer needle [18,19,20]. A stronger immune response and lower incidence of adverse events were reported in female elderly immunized with influenza and 23-valent pneumococcal vaccines intramuscularly, compared with immunized subcutaneously [21,22]. Recently, a significant immune response was reported among the Japanese elderly immunized with the subunit herpes zoster vaccine, and a lower incidence of adverse events was noted [23,24]. Lower incidences of local reactions, redness, swelling, induration, and pain were reported in infants and the elderly through IM administration of vaccines [17,18,19,20,21,22,23,24].

An appropriate injection procedure is essential for safe immunization. It depends on the needle size and length. The appropriate needle length for intramuscular administration is the total length of the epidermis, subcutaneous tissues, and muscle mass without reaching the bone surface. This varies depending on age, gender, anatomical differences, and different nursing care levels [11,12,13]. CDC recommended a 25 mm (1-inch) needle length [25]. In our previous study, the appropriate needle length for intramuscular injections in Japanese infants was 16 mm (5/8-inch) at any age and sites of lateral quadriceps for young infants and deltoid for those aged > 1 year, with 25 mm (1-inch) needles at a 90° angle being associated with the risk of overpenetration [14]. In the present study, ultrasonic echography was performed for subjects ≥ 50 years of age in outpatient clinics and facilities of nursing homes for the elderly. The epidermis is thicker in males than in females in the elderly aged 50–74 years. Subcutaneous mass was larger in females than males, but there was no gender difference in the muscle volume in those aged 50–74 years. Regarding the subcutaneous volume, the depth from the skin surface to muscle fascia was greater in females than in males aged 50–74 years. We examined the subcutaneous volume depending on the different nursing care levels in subjects aged ≥ 75 years. There was no significant difference among them. Irrespective of gender, age, or the degree of nursing care, 16 mm (five-eighths of an inch) was an appropriate length to administer vaccines into muscle tissues while avoiding the bone.

As a limitation of the study, the depth from the skin surface to muscle fascia differs individually. An appropriate needle length for most Japanese elderly is 16 mm. Three subjects in the 50–64-year group were outliers (16.7, 18.1, and 16.9 mm), shown in Figure 2; for example, a 51-year-old male, 179.3-cm height, body weight of 151.7 kg, BMI of 47.2, and upper arm circumference of 56.5 cm. The depth from the skin surface to muscle fascia was longer than 16 mm in three subjects with a common feature of large subcutaneous fatty mass volume, 14.5, 15.5, and 15.2 mm. The appropriate needle length was not 16 mm in these subjects. Therefore, a longer needle is required for those with a larger size.

## 5. Conclusions

An appropriate needle length for IM administration is 16 mm five-eighths of an inch) for Japanese elderly aged ≥ 50 years with an average physique. The usage of a longer needle may be considered for the elderly with an extremely large constitution.

## Figures and Tables

**Figure 1 healthcare-10-00800-f001:**
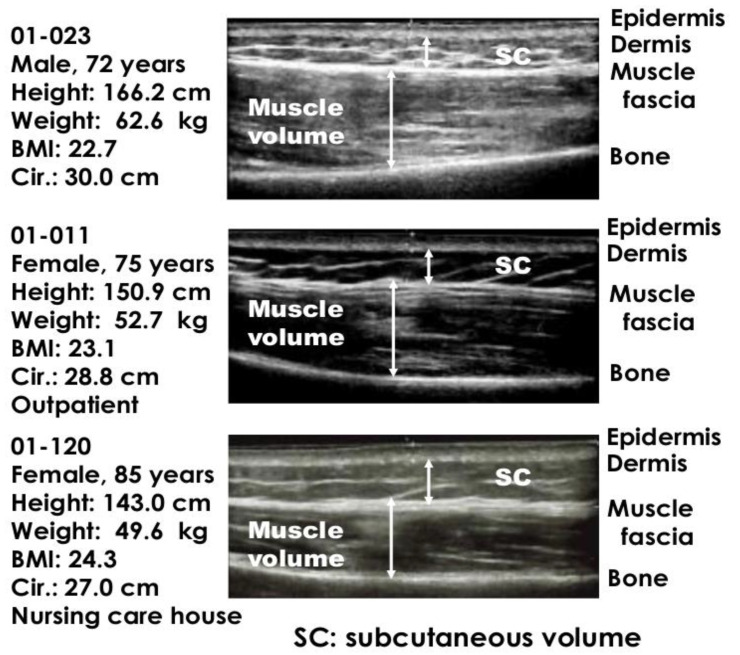
Typical echography in different age groups in the center of the deltoid muscle area.

**Figure 2 healthcare-10-00800-f002:**
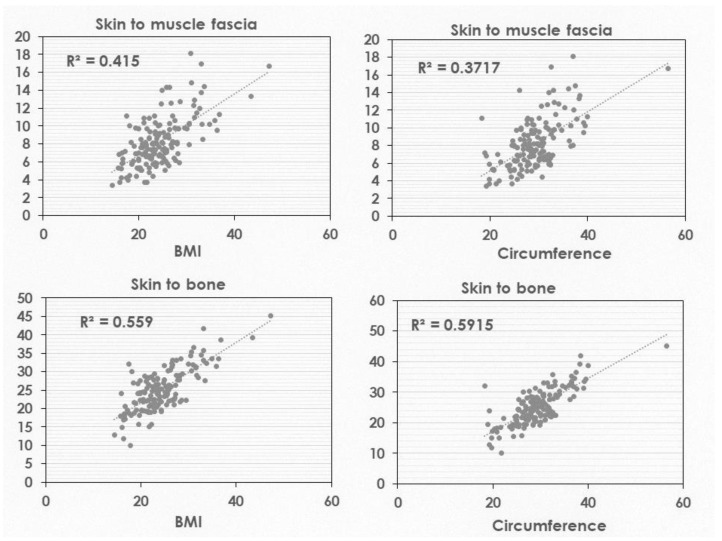
Correlation of BMI and circumference through the deltoid area with depth from the skin surface to muscle fascia and bone.

**Table 1 healthcare-10-00800-t001:** Number of subjects in separate groups.

Category	Male	Female
50–64 years (n = 50)	26	24
65–74 years (n = 50)	25	25
≥75 years		
Outpatients (n = 20)	14	6
Needed nursing care (n = 18)	8	10
Nursing home bed-ridden (n = 18)	8	10
Total (n = 156)	81	75

**Table 2 healthcare-10-00800-t002:** Body size background (height, body weight, BMI, and circumference of upper arm).

	Height (cm)	Body Weight (kg)	BMI	Circumference (cm)
50–64 years (n = 50)	164.34 ± 8.86 **	73.32 ± 20.44 **	26.96 ± 6.40 *	32.09 ± 5.59 *
65–74 years (n = 50)	159.60 ± 9.05 *	63.33 ± 10.34 **	24.81 ± 3.79 **	30.02 ± 3.20 **
≥75 years (n = 38)	155.34 ± 9.66	54.95 ± 10.34 **	22.64 ± 2.97 **	27.30 ± 3.28 **
Nursing home (n = 18)	154.72 ± 8.20	44.64 ± 9.99	18.46 ± 2.72	22.90 ± 3.66

*: *p* < 0.05, **: *p* < 0.01.

**Table 3 healthcare-10-00800-t003:** Depth from the skin surface to muscle fascia and that to the bone.

Age Group	Epidermis	SC Volume	Muscle Volume	Skin Surface to Muscle Fascia	Skin Surface to Bone
50–64 years (n = 50)	2.26 ± 0.54	6.90 ± 3.01	19.44 ± 4.44	9.16 ± 3.02	28.60 ± 5.91
≥75 years (n = 38)	1.93 ± 0.54 **	5.58 ± 2.03 *	15.02 ± 3.15 **	7.52 ± 2.13 **	22.54 ± 3.85 **
Outpatients (n = 20)	1.87 ± 0.46	5.65 ± 1.60	15.31 ± 3.48	7.51 ± 1.72	22.82 ± 4.06 *
need nursing care (n = 18)	2.01 ± 0.61	5.52 ± 2.46	14.70 ± 2.80	7.53 ± 2.56	22.23 ± 3.68
Bedridden (n = 18)	1.71 ± 0.39	4.73 ± 1.79	13.03 ± 3.75	6.44 ± 2.05	19.47 ± 5.55 *

*: *p* < 0.05, **: *p* < 0.01, SC: subcutaneous.

**Table 4 healthcare-10-00800-t004:** Gender difference in subcutaneous mass.

	Subcutaneous Mass in Subjects 50–64 Years				
	Epidermis	Subcutaneous Volume	Muscle Volume	Skin Surface to Muscle Fascia	Skin Surface to Bone
Male (n = 26)	2.51 ± 0.55 **	5.62 ± 2.80	20.23 ± 4.21	8.12 ± 2.94	28.36 ± 5.62
Female (n = 24)	2.00 ± 0.39	8.29 ± 2.63 **	18.58 ± 4.60	10.29 ± 2.74 **	28.87 ± 6.33
	**Subcutaneous Mass in subjects 65–74 Years**				
Male (n = 26)	2.27 ± 0.43 **	5.10 ± 1.55	16.82 ± 2.29	7.37 ± 1.75	24.19 ± 2.76
Female (n = 24)	1.83 ± 0.37	7.72 ± 3.06 **	17.09 ± 3.94	9.55 ± 3.20 **	26.64 ± 5.10

**: *p* < 0.01, *: *p* < 0.05.

## Data Availability

Data set is stored in Laboratory of Viral Infection, Ömura Satoshi Memorial Institute, Kitasato University.

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
