# Peer review of "Appropriate Needle Length Determined by Ultrasonic Echography for Intramuscular Injection in Japanese Elderly over 50 Years"

_healthcare, 2022, doi:10.3390/healthcare10050800_

Round 1
Reviewer 1 Report
The purpose of this manuscript was to investigate appropriate needle length determined by ultrasonic echography for performing appropriate intramuscular injection in Japanese elderly over 50 years. In terms of the purpose, it is an important issue or not for vaccine injection might not be the same in different countries. In addition, the confounding variables didn’t consider in statistical analysis. The inferential statistical techniques used were not comprehensive. As a result, advanced validation needs to be conducted for the future clinical applications.
Author Response
Reviewer 1.
Thank you for your valuable comments. Followings are my response.
The purpose of this manuscript was to investigate appropriate needle length determined by ultrasonic echography for performing appropriate intramuscular injection in Japanese elderly over 50 years. In terms of the purpose, it is an important issue or not for vaccine injection might not be the same in different countries. In addition, the confounding variables didn’t consider in statistical analysis. The inferential statistical techniques used were not comprehensive. As a result, advanced validation needs to be conducted for the future clinical applications.
【Response】I understood the comment. Subjects were enrolled randomly who visited the clinics in the study, and therefore there were many confounding factors: history of lifestyle, history of chronic diseases, and dietary habits. Especially in subjects ≥75years, their daily activity was quite different from each other and were divided into three groups; outpatient, needed nursing care, and bedridden. Groups of 50-64 years and 60-70 years were subjects who were regularly visiting our outpatient clinics. It would be hard to consider the confounding factors. I want to know the anatomical structure of deltoid muscle area. Limitation of the study is for the Japanese elderly, and the last sentence was changed as following, Line 25-27:
Our study found that 5/8-inch (16 mm) needle was appropriate length for average-sized elderly aged ≥50 years, but it should be longer for those with a large body size.
Reviewer 2 Report
Dear authors,
Thank you for the opportunity to review your manuscript. The topic of the appropriate length for intramuscular injection though not entirely novel, was interesting because I was unaware that in Japan these injections are given subcutaneously because of prior experiences of muscular contraction in 1960-70s. Considering that intramuscular injection is a common feature in the health system here in North America, makes your manuscript intriguing to me as a reviewer. Also it speaks to the importance of revising clinical care to generate new evidence for improve patient care.
Abstract - You say "all Japanese". I would caution the authors about making such a generalized statement that can fuel scientific racism. Please consider revising to something like.. our study found 16mm needle was appropriate length for elderly aged >50 but it should....
Line 33 -"This is because of muscular contraction in the 1960s to 1970s after the administration of antibiotics together with antipyretics in thigh lesions"; this sentence is unclear. Do you mean the administration led to lesion in the thigh?
Line 46 - HPV is recommended as IM injection in other parts of the world not just Japan; consider revising.
Line 65 - Please consider revising the use of ethnicity? are you talking about ethnicity here or anatomical size? Scientists have been arguing for consistency of language. Here your discussion is on physical characteristics of people, which ancestral origin, anatomical size might be closer to what you are studying vs ethnicity which is customs, religion and values etc of groups of people. I suggest you use a word that is more descriptive of your research objective. Read this: https://www.scientificamerican.com/article/race-is-a-social-construct-scientists-argue/
Line 72 - describe to the reader how you arrived at the sample size.
Line 75 - is there a reason for oversampling in the >75 age group?
Line 77 - You use the word "quality"; do you mean functional status so activity of daily living? It is not directly clear how quality relates to your research objective but functional status make sense.
Line 83 - You can simply put the table into parentheses at the end of sentence in line 82
Line 86 - grammar; suggest - "the sample represents the average..."
Line 89 - how do you define physique?
Line 94 - spelling error of "ethics"
Line 115 - Sentence structure - consider revising
Line 116-122 - consider revising this section to be consistent, parallel structure in line 122-123,
Table 2 - put asterisks next to the values not on separate line. Also unclear why is the nursing group is included here? I would assume this group would be included with the other functional groups - outpatient, bedridden?
Line 129-132 - unclear; is the echograms have different age group compare to your overall study?
Line 134 - 139 - consider revising for consistency ad parallel structure (mean/SD for each group with associated pvalue)
Line 152 - remove and consider my prior suggestion.
Table 4 - ensure words on same level - eg subcutaneous
Line 160-161 - revise: Irrespective of the background of age, body structure, and sex differences, a 16-mm 160 needle reached the muscular mass, and a 25-mm needle reached the bone in some slim 161 elderly subjects.
Line 164 - revise: The presence of a correlation was investigated between the distance from the skin ...
Line 167 - Last part of the sentence is unclear. Do you mean there is no correlation between distance to the bone and circumference?
Line 192 - Consider "this" instead of it
Line 192 - "varies depending on the age, sex, racial differences, and different nursing" see my previous comment about scientific racism; you are talking about anatomic differences; not race or racism.
Line 193 - include citation for WHO reference
Line 213 - "enormous size" consider another sensitivity terminology -- larger size or bigger anatomy
Line 215 - consider: IM administration is 16 mm (5/8 inch) for Japanese 215 elderly aged ≥50 years with an average physique.
Overall the manuscript needs revision with attention to sentence structure and clarity. Thank you for the opportunity and hope the feedback is helpful.
Author Response
Reviewer 2
Thank you for your valuable and detailed comments. Followings are my response.
Dear authors,
Thank you for the opportunity to review your manuscript. The topic of the appropriate length for intramuscular injection though not entirely novel, was interesting because I was unaware that in Japan these injections are given subcutaneously because of prior experiences of muscular contraction in 1960-70s. Considering that intramuscular injection is a common feature in the health system here in North America, makes your manuscript intriguing to me as a reviewer. Also, it speaks to the importance of revising clinical care to generate new evidence for improve patient care.
Abstract - You say, "all Japanese". I would caution the authors about making such a generalized statement that can fuel scientific racism. Please consider revising to something like. our study found 16mm needle was appropriate length for elderly aged >50 but it should.... 【Response】I understood and the last sentence was changed as recommended, Line 25-27; Our study found that 5/8-inch (16 mm) needle was appropriate length for average-sized elderly aged ≥50 years, but it should be longer for those with a large body size.
Line 46 - HPV is recommended as IM injection in other parts of the world not just Japan; consider revising. 【Response】HPV was recommended through IM by package insert and Ministry of Health, Labour, and Welfare. But IM procedure was unfamiliar to general physicians and then increased number of adverse reactions was reported. I added the phase as following, Line 45-48: Newly introduced human papilloma virus vaccine (HPV) was recommended IM from the beginning of introduction to Japanese female middle school students [6]. Some local pains may be due to the unfamiliar procedure of IM administration.
Line 65 - Please consider revising the use of ethnicity? are you talking about ethnicity here or anatomical size? Scientists have been arguing for consistency of language. Here your discussion is on physical characteristics of people, which ancestral origin, anatomical size might be closer to what you are studying vs ethnicity which is customs, religion, and values etc. of groups of people. I suggest you use a word that is more descriptive of your research objective. Read this: https://www.scientificamerican.com/article/race-is-a-social-construct-scientists-argue/ 【Response】Line 66: “Ethnicity” was changed to ” racial differences and body size”.
Line 72 - describe to the reader how you arrived at the sample size. 【Response】Preceding reports, enrolled number was 20-30 subjects for each aged group for 50-59, 60-69 ……
Line 75 - is there a reason for oversampling in the >75 age group? 【Response】≥75years, their daily activity was quite different from each other and were divided into three groups; outpatients, needed nursing care, and bedridden. 20 subjects for each were enrolled initially.
Line 77 - You use the word "quality"; do you mean functional status so activity of daily living? It is not directly clear how quality relates to your research objective, but functional status make sense. 【Response】Line 78: This word was changed to activity of daily living
Line 83 - You can simply put the table into parentheses at the end of sentence in line 82 【Response】This sentence was changed as following, Line 82-83. Two cases were excluded each in groups aged ≥75 years with nursing care and bedridden because of blurred echography, and so a total of 156 subjects were included (Table 1).
Line 86 - grammar; suggest - "the sample represents the average..." 【Response】It was changed as commented, Line 86. This sample size represents of the average physique of elderly and appropriate sampling.
Line 89 - how do you define physique? 【Response】It was mentioned in the following section 2.2. Study design. Line 91-91. Before the entry of the study, height and body weight were measured, and body mass index (BMI) was calculated.
Line 94 - spelling error of "ethics" 【Response】It was revised, Line 95.
Line 116-122 - consider revising this section to be consistent, parallel structure in line 122-123, 【Response】This sentence was revised as following, Line 116-118: Height was 164.34 ± 8.86 cm (mean ± 1.0 SD) in subjects aged 50-64 years higher than that in those aged 65-74 years (p<0.01), being higher (p<0.05) compared with that in those aged ≥75 years.
Table 2 - put asterisks next to the values not on separate line. Also unclear why is the nursing group is included here? I would assume this group would be included with the other functional groups - outpatient, bedridden? 【Response】Statistical analysis was performed between the two vertically aligned cells. In subjects ≥75years, their daily activity was quite different from each other and were divided into three groups; outpatient, needed nursing care, and bedridden. Groups 50-64 years and 60-70 years were average subjects who were regularly visiting our outpatient clinics.
Line 129-132 - unclear; is the echograms have different age group compared to your overall study? 【Response】I added the following words as following, Line 130: Ultrasonic echogram was taken from all participants and typical echograms are shown in different age groups:
Line 134 - 139 - consider revising for consistency ad parallel structure (mean/SD for each group with associate p value). 【Response】P values were moved as following, Line 136-141: The epidermis is thin at 1.93 ± 0.54 mm (mean ± 1.0 SD, p<0.01) in those aged ≥75 years, compared with those aged 50-64 years. Subcutaneous volume length was 5.58 ±2.03 mm (mean ± 1.0 SD, p<0.05) in those aged ≥75 years, being shorter than that in those aged 50-64 years, and the muscle volume length was 15.02 ± 3.15 mm (mean ± 1.0 SD, p<0.01) in those aged ≥75 years and 16.95 ± 3.19 mm (mean ± 1.0 SD, p<0.01) in those 65-74 years, being significantly shorter than 19.44 ± 4.44 mm in those aged 50-64 years.
Line 152 - remove and consider my prior suggestion. 【Response】It was changed to “gender”, Line 153.
Table 4 - ensure words on same level - e.g., subcutaneous. 【Response】Table 4 was revised.
Line 160-161 - revise: Irrespective of the background of age, body structure, and sex differences, a 16-mm 160 needle reached the muscular mass, and a 25-mm needle reached the bone in some slim 161 elderly subjects. 【Response】The sentence was revised as commented, Line 162-163.
Line 167 - Last part of the sentence is unclear. Do you mean there is no correlation between distance to the bone and circumference? 【Response】The word not was missed, and this sentence was revised as following, Line 167-168: Depth from the skin surface to muscle fascia was not correlated with BMI or circumference but that to bone was correlated with the circumference around the deltoid area.
Line 192 - Consider "this" instead of it.
【Response】It was corrected, Line 191.
Line 192 - "varies depending on the age, sex, racial differences, and different nursing" see my previous comment about scientific racism; you are talking about anatomic differences; not race or racism. 【Response】It was changed, Line 193-193. This varies depending on the age, gender, anatomical differences, and different nursing care levels [11-13].
Line 193 - include citation for WHO reference.
【Response】Home page of CDC is easily accessible and referred as Ref. No. 25, Line 193.
- CDC. Vaccine administration: intramuscular (IM) injection adults 19 years age and older. Vaccine Administration: Intramuscular (IM) injections: Adults 19 years of age and older (cdc.gov)
Line 213 - "enormous size" consider another sensitivity terminology -- larger size or bigger anatomy. 【Response】Revised as commented, Line 215.
Line 215 - consider: IM administration is 16 mm (5/8 inch) for Japanese 215 elderly aged ≥50 years with an average physique. 【Response】Line 217, I want to keep Japanese in this sentence because of the title of this paper.
Overall, the manuscript needs revision with attention to sentence structure and clarity. Thank you for the opportunity and hope the feedback is helpful.
Reviewer 3 Report
I would like to read more about the background of IM and SC administrations in the country.
Well done and relevant work.
I would recommend that in the method was clarify the way as the sample was selected (by convenience? With what criteria? In the selection process, you detect any problem in selecting people, ie dementia? )
It would be beneficial for the work to include perceived limitations and limitations associated with generalizing the conclusions (given the comparison with the global population)
Author Response
Reviewer 3
Thank you for your valuable and detailed comments. Followings are my response.
I would like to read more about the background of IM and SC administrations in the country. 【Response】 As mentioned in the text, Line 33-39. In the 1960s to 1970s, it was noticed as medical malpractice that the administration of antibiotics together with antipyretics in the thigh lesion led the muscle contraction in young infants [3]. The reason for the contraction was due to extremely high osmotic pressure and pH different from the physiological condition [3,4]. The reason for the contraction was due to extremely high osmotic pressure and pH different from the physiological condition [3,4]. Although there was no case of muscle contracture after the administration of vaccines in children, vaccines have been administered through SC ever since IM injection was prohibited in 1976.
Well done and relevant work.
I would recommend that in the method was clarify the way as the sample was selected (by convenience?) With what criteria? In the selection process, you detect any problem in selecting people, ie dementia? 【Response】 In the section of subjects. Patient with Dementia was not included in this study. Four subjects were poor mutual understanding and accurate echography could not obtain. Exclusion criteria were subjects with anomalies in upper limbs, peripheral nervous disorders, muscular illness of upper limbs, acute systematic illnesses, and serious chronic ailments. Two cases were excluded each in groups aged ≥75 years with nursing care and bedridden because of blurred echography
Reviewer 4 Report
The study of Nakayama et al focuses on needle length for intramuscular injection in Japanese patients. The topic is very interesting, the paper is well structured and nicely written.
I just want to address a minor comment:
- The study focuses on Japanese patients only. How would the authors interprete the results with respect to other ethnic groups (e.g. more/less fat, different needle length). Please discuss!
Author Response
The study focuses on Japanese patients only. How would the authors interprete the results with respect to other ethnic groups , please discuss
[Response]The following paragraph is added in the revised version in the very last paragraph, Line 209-214. Three subjects in the 50-64 year group were outliers, shown in Figure2: for example, 51-year-old male, 179.3 cm height, body weight of 151.7 kg, BMI of 47.2, and upper arm circumference of 56.5 cm. The depth from the skin surface to muscle fascia was longer than 16 mm in three subjects who had a common feature of higher subcutaneous mass volume, 14.5, 15.5, and 15.2 mm.